# The Multiple Mitotic Roles of the ASPM Orthologous Proteins: Insight into the Etiology of ASPM-Dependent Microcephaly

**DOI:** 10.3390/cells12060922

**Published:** 2023-03-16

**Authors:** Alyona V. Razuvaeva, Lucia Graziadio, Valeria Palumbo, Gera A. Pavlova, Julia V. Popova, Alexey V. Pindyurin, Silvia Bonaccorsi, Maria Patrizia Somma, Maurizio Gatti

**Affiliations:** 1Department of Regulation of Genetic Processes, Institute of Molecular and Cellular Biology, Siberian Branch of Russian Academy of Sciences, Novosibirsk 630090, Russia; alena.razuvaeva@mcb.nsc.ru (A.V.R.); popova@mcb.nsc.ru (J.V.P.); aleksey.pindyurin@gmail.com (A.V.P.); 2Institute of Cytology and Genetics, Siberian Branch of Russian Academy of Sciences, Novosibirsk 630090, Russia; 3IBPM CNR c/o Department of Biology and Biotechnology, Sapienza University of Rome, 00185 Rome, Italy; lucia.graziadio1@gmail.com; 4Department of Biology and Biotechnology, Sapienza University of Rome, 00185 Rome, Italy; valeria.palumbo@uniroma1.it (V.P.); silvia.bonaccorsi@fondazione.uniroma1.it (S.B.); 5Wellcome Centre for Cell Biology, School of Biological Sciences, University of Edinburgh, Edinburgh EH9 3BF, UK; gpavlova@exseed.ed.ac.uk; 6Laboratory of Bioengineering, Novosibirsk State Agrarian University, Novosibirsk 630039, Russia

**Keywords:** *Drosophila* Asp, mouse Aspm, human ASPM, microcephaly, spindle poles, asters, central spindle, DNA replication, DNA repair, cell cycle progression

## Abstract

The *Drosophila abnormal spindle* (*asp*) gene was discovered about 40 years ago and shown to be required for both mitotic and meiotic cell division. Subsequent studies showed that *asp* is highly conserved and that mutations in its human ortholog *ASPM* (*Abnormal Spindle-like Microcephaly-associated*; or *MCPH5*) are the most common cause of autosomal recessive primary microcephaly. This finding greatly stimulated research on *ASPM* and its fly and mouse (*Aspm*) orthologs. The three Asp orthologous proteins bind the microtubules (MTs) minus ends during cell division and also function in interphase nuclei. Investigations on different cell types showed that Asp/Aspm/ASPM depletion disrupts one or more of the following mitotic processes: aster formation, spindle pole focusing, centrosome-spindle coupling, spindle orientation, metaphase-to-anaphase progression, chromosome segregation, and cytokinesis. In addition, ASPM physically interacts with components of the DNA repair and replication machineries and is required for the maintenance of chromosomal DNA stability. We propose the working hypothesis that the *asp*/*Aspm*/*ASPM* genes play the same conserved functions in *Drosophila*, mouse, and human cells. Human microcephaly is a genetically heterogeneous disorder caused by mutations in 30 different genes that play a variety of functions required for cell division and chromosomal DNA integrity. Our hypothesis postulates that *ASPM* recapitulates the functions of most human microcephaly genes and provides a justification for why *ASPM* is the most frequently mutated gene in autosomal recessive primary microcephaly.

## 1. Introduction

*abnormal spindle* (*asp*) was one of the first mitotic genes discovered in *Drosophila asp* mutants were shown to exhibit a strong mitotic phenotype in larval brain cells and morphologically abnormal spindles in male meiosis [1]. Molecular cloning of *asp* revealed that the gene encodes a large microtubule (MT)-binding protein that accumulates at the spindle poles [2]. The interest in *asp* was greatly heightened by the discovery that mutations in its human ortholog *ASPM* (*Abnormal Spindle-like Microcephaly-associated*) are the most common cause of autosomal recessive primary microcephaly (MCPH) [3,4,5]. MCPH is a rare and genetically heterogeneous disorder characterized by reduced head circumference of up to one-third of the normal volume at birth, resulting in mild-to-moderate intellectual disability. The principal cause of MCPH is a decreased production of neurons in the developing neocortex due to defects in progenitor cell proliferation and/or apoptosis (reviewed by [6,7,8]). To date (February 2023), 30 *MCPH* genes have been identified that are listed in the Online Mendelian Inheritance in Man (OMIM) database (https://omim.org; accessed on 15 December 2022), with *ASPM* designated as *MCPH5*. Most of these genes are involved in different aspects of mitotic division, including centriole biogenesis, centrosome-driven MT nucleation, kinetochore assembly and function, and spindle formation. However, a fraction of them are required for chromatin condensation and remodeling, DNA repair, and chromosome stability [7,8]. Remarkably, mutations in *ASPM* are responsible for more than 40% of MCPH cases [5].

Another reason for interest in the *ASPM* gene is its role in tumor development. *ASPM* expression is upregulated in several cancers, including prostate cancer, glioblastoma, and hepatocellular carcinoma, and increased *ASPM* expression is associated with tumor progression and poor clinical prognosis. It has also been shown that siRNA-mediated ASPM depletion strongly inhibits tumor cell proliferation ([9,10,11] and references therein). These and other studies suggested considering *ASPM* and the other *MCPH* genes as promising therapeutic targets in brain tumors (reviewed by [12,13]).

Although the Asp protein and its human (ASPM) and mouse (Aspm) orthologs have been studied for many years, a precise functional comparison between the three proteins has never been made. Here, we review the studies on the mitotic roles of these proteins in different *Drosophila*, mouse, and human cell types. These studies reveal that Asp, Aspm, and ASPM tend to have cell type-specific requirements. However, despite this fact and some apparently conflicting results, we propose a model that suggests a common mitotic function for the three orthologous proteins. We believe that this model explains most of the phenotypes associated with *asp* and *ASPM* mutations, providing a basis for future studies on the conserved *asp*/*Aspm*/*ASPM* gene.

## 2. The Structure of the Orthologous Asp/Aspm/ASPM Proteins

The Asp/Aspm/ASPM proteins are highly conserved; the NCBI database (https://www.ncbi.nlm.nih.gov; accessed on 20 December 2022) contains more than 450 orthologs of these proteins. Here we will focus on *Drosophila* Asp, mouse Aspm, and human ASPM, which are functionally best characterized. When we refer to one of the three orthologs, we will use its specific gene/protein name. To refer to the three orthologs together, we will indicate them as Asp/ASPM proteins. The three proteins have different lengths, although they contain the same structural domains. The difference in size (Asp, 1954 aa; Aspm 3122 aa; ASPM 3477 aa) is mainly due to the number of isoleucine-glutamine (IQ) motifs that varies from 24 in *Drosophila* to 60 and 81 in mice and humans, respectively (Figure 1) [3,14,15,16]. The IQ motifs of *Drosophila*, *C. elegans*, mouse and human proteins bind calmodulin (CaM) ([17,18,19,20]; see Section 5 below for Asp/ASPM-CaM interactions).

The N terminal region of the Asp/ASPM proteins contains a Hydin or ASH (ASPM, SPD-2, Hydin) domain with a putative MT-binding function [16]. It has been reported that the N terminus of Asp/ASPM also contains a major sperm protein (MSP) domain that overlaps the Hydin domain [19]. Besides the multiple IQ motifs, the central part of the Asp/ASPM proteins contains a variable number of calponin homology (CH) domains, which also have an MT binding and bundling function [19,21]. The C terminal region of the Asp/ASPM orthologs is likely to contain Heat/Armadillo-like repeats, whose function is currently unknown (Figure 1).

Although the structural features of the Asp/ASPM proteins suggest that they might be interchangeable, we have very little data addressing this issue. We know that the expression of human *ASPM* transgene in *Aspm* mutant mice can rescue all mutant phenotypes, including the reduction in brain size and the massive loss of germ cells in both males and females [22]. However, whether *asp* and *Aspm* can rescue the mutant phenotypes elicited by *ASPM* mutation has never been investigated.

## 3. Subcellular Localization of the Asp/ASPM Proteins

Early studies using antibodies against an Asp N terminal fragment of 512 amino acids showed that Asp localizes to the polar regions of the spindles and to the telophase central spindle in syncytial *Drosophila* embryos [2]. Subsequent studies expanded and refined these initial observations showing that Asp accumulates at the spindles poles of larval neuroblasts [23,24], epithelial cells [25], S2 tissue culture cells [18,21,26,27], and meiotic cells of both males and females [24,28,29]. Specifically, it has been reported that Asp accumulates at the transition region between the spindle and the centrosome, with Asp immunostaining partially overlapping the immunofluorescence signal elicited by γ-tubulin or Centrosomin (Cnn, the ortholog of the human centrosomal protein CDK5RAP2). However, a series of observations indicate that Asp localizes to the spindle poles independently of the centrosomes and that it is not an integral component of the centrosome. Namely, (i) Asp accumulates to the spindle poles of larval brain cells devoid of functional centrosomes, such as those of mutants in *asterless* (*asl*; encoding the ortholog of human CEP152), *cnn* or *dd4* (encoding a γ-tubulin ring component) [21,24,30], (ii) the γ-tubulin and Cnn signals in *asp* mutant metaphases are of similar intensity to those of controls [24], and, most importantly, (iii) in colchicine-treated embryos and S2 cells with fully depolymerized spindle MTs Asp does not localize to the centrosome [24,27]. 

In addition to the spindle poles, Asp accumulates at the central spindle. The central spindle, or intercellular bridge, is a prominent MT bundle that forms during late telophase. It consists of antiparallel MTs with the plus ends interdigitating at the center of the bundle, where they associate with many different proteins generating a discrete structure currently called the midbody. These proteins form a dense cluster (midbody ring) that impedes the access of anti-tubulin antibodies, resulting in a dark zone after tubulin immunostaining [31]. Asp associates with the sides of the MT bundle that face the telophase nuclei. This peculiar localization has been observed in different cell types, including larval neuroblasts, S2 tissue culture cells, and male meiotic cells [24,27,28]. Asp also accumulates at the extremities of the central spindle of embryonic telophases. In these syncytial divisions, where cytokinesis does not occur, the central spindle is not hourglass-shaped as in cells with a contractile ring; it is instead diamond-shaped with sharply focused extremities that are highly enriched in Asp [24]. The localization of Asp at the spindle poles and at the outer sides of the central spindle suggests that Asp binds and crosslinks the minus ends of the spindle MTs [21,24,26,28]. This suggestion was recently corroborated by in vitro studies showing that Asp accumulates at the MT minus ends [19].

Besides its accumulation at the spindle poles and central spindle extremities, antibody staining revealed a weak Asp signal along the spindle MTs of different cell types, including larval neuroblasts, spermatocytes, and S2 cells [24,26,28]. In addition, the prometaphase and metaphase spindles of live larval neuroblasts and S2 cells, both expressing Asp-GFP, displayed discrete fluorescent signals along the spindle MTs. Imaging of these Asp-GFP particles revealed that they stream towards the spindle poles and are eventually incorporated into the polar Asp pool [18,21]. A poleward flow of Asp-GFP particles was also observed in the epithelial cell spindles of *Drosophila* pupal notum [25].

Aspm localization in neuroepithelial (NE) mouse cells is very similar to the Asp localization in *Drosophila* cells. Aspm localizes to the spindle poles throughout mitosis, accumulating in the immediate vicinity of the γ-tubulin signal of centrosomes. Aspm staining does not overlap the γ-tubulin immunoreactivity, and Aspm is absent from centrosomes during interphase [32]. In addition, Aspm localizes to the outer regions of the central spindle, but it is excluded from the midzone/midbody [22]. Thus, like its *Drosophila* ortholog, mouse Aspm appears to localize to spindle regions enriched in MT minus ends, consistent with the observation that Aspm preferentially associates with the MT minus ends in vitro [19].

Early studies on U2OS tissue culture cells suggested that human ASPM co-localizes with the centrosomes at the spindle poles and is enriched at the centrosomes in interphase nuclei [33]. Similar conclusions were reached in analyses performed in HeLa cells [34]. However, subsequent studies in U2OS cells showed that ASPM forms a ring around the centrosomes at the spindle poles, with little overlap with the γ-tubulin staining [35]. These studies did not provide evidence for ASPM localization at the interphase centrosomes. They also showed that after tubulin depolymerization with nocodazole, ASPM immunostaining at the spindle poles is lost, while γ-tubulin staining is unaffected [35]. However, in another investigation, antibody staining showed discrete ASPM signals located next to centrin (a centriolar marker) signals in interphase cells [36]. Thus, while it is clear that ASPM localization at the spindle poles is MT-dependent and centrosome-independent, the relationships between ASPM and the centrosome are not fully clarified.

ASPM localizes to the central spindle like its *Drosophila* and mouse counterparts, but there are some conflicting results about its precise localization to this structure. Using a commercial anti-human ASPM antibody and an antibody raised against amino acids 1-418 of rat ASPM, Paramasivam et al. [34] showed that both antibodies stain the midbody (dark zone) of HeLa cells. In a subsequent study, Higgins et al. [35] showed that an antibody directed to an ASPM N terminal peptide (aa 363–386) associates with the extremities of the central spindle but not with the midbody. In contrast, the same study showed that another antibody raised against a C terminal peptide (aa 3443–3458) of ASPM specifically decorates the midbody but fails to stain the lateral regions of the central spindle. This discrepancy was not addressed, and the authors focused on the relationships between ASPM and the centrosomes [35]. 

More recently, Jiang et al. [19] analyzed the properties and the mitotic behavior of a complete GFP-ASPM protein generated by the insertion of a GFP coding sequence just upstream of the first *ASPM* exon. In HeLa cells, GFP-ASPM accumulated to the spindle poles throughout mitosis; it also accumulated at central spindle extremities but was absent from the midbody. Importantly, Jiang et al. [19] also showed that ASPM binds the MT minus ends both in vitro and in living cells.

Collectively the extant results indicate that the orthologous Asp/ASPM proteins exhibit very similar, if not identical, localization patterns. They do not appear to be integral centrosome components but accumulate to the spindle poles, where they are likely to bind and crosslink the minus ends of the MTs that detach from the centrosomes ([37]; reviewed by [38]). In addition, Asp and its orthologues accumulate at the extremities of the central spindle that are enriched in MT minus ends. It is unlikely that they are also part of the midbody, as this structure is thought to contain little to no MT minus ends [31,39,40]. The detection of ASPM at the midbody with some specific antibodies might reflect a cross-reaction with one of the many proteins that compose this structure [41]. 

## 4. The Mitotic Functions of the *asp*/*ASPM* Genes

Strong *Drosophila* mutations such as *asp^1^* and *asp^E3^* are semilethal. Less than 20% of the homozygous progeny from heterozygous mothers survive to adulthood, and most of them have severely reduced head size compared to the wild type. Homozygous males are fertile but produce frequent aneuploid gametes due to meiotic nondisjunction, while females are sterile [1,16,29,42]. In mice, strong and potentially null mutations in *Aspm* are not lethal but reduce brain size and cause a drastic loss of germ cells in both testes and ovaries, leading to reduced fertility [22,43]. Mutations leading to different and extensive truncations of the human ASPM protein only cause a severe reduction in brain size with no apparent non-neurological phenotypes [3,4]. A series of studies carried out in a variety of systems have attributed several functions to the Asp orthologues at both the cellular and subcellular levels. Below we summarize these functions and attempt to define their degree of conservation between cells of the same species and among fly and mammalian cells. 

### 4.1. The Role of Asp/ASPM Orthologs in Mitotic Progression

The early studies on *Drosophila asp* revealed that this gene is required for mitotic progression. The brains of *asp* mutant larvae displayed a high mitotic index (MI) with frequent metaphase-like figures and almost no anaphases. In addition, they showed frequent hyperploid and polyploid cells [1,44,45]. The metaphase arrest phenotype is probably due to the activity of the spindle assembly checkpoint (SAC). Indeed, while *asp* mutant brains exhibit a MI three- to six-fold higher than wild-type brains, in *asp* double mutants with either *rod* or *mad2*, which encode SAC components, the MI drops to the wild-type level [46]. This finding raises a question about the mechanisms that trigger the SAC response in *asp* mutant cells. It has been proposed that SAC signaling might be caused by centrosome misplacement that would influence the dynamic interaction between MTs and kinetochores ([18]; see also Section 4.3 below). Another possibility is that Asp has a role in kinetochore-driven MT growth and that, in the absence of Asp, irregularities in this process trigger the SAC ([27]; see also Section 4.6 below). Whatever the mechanism of SAC triggering, the polyploid cells observed in *asp* mutant brains are likely to be a consequence of checkpoint adaptation or mitotic slippage, a process through which a cell arrested in metaphase by the SAC exits mitosis and enters next G1 as a tetraploid cell (reviewed in [47]). Although the data suggest that the polyploid cells observed in *asp* mutant brains are due to mitotic slippage, they do not exclude a role of Asp in cytokinesis. As larval cells of *asp* mutant brains do not undergo anaphase and telophase, the role of Asp in cytokinesis cannot be assessed. However, an involvement of Asp and its orthologues in cytokinesis has been demonstrated in other cell types (see Section 4.5 below).

Besides brain cells, an increased MI and a metaphase delay phenotype following Asp depletion have been observed in S2 cells [26,48] but not in other *Drosophila* somatic cells. However, an analysis of cell division in live NE cells (that are distinct from central brain cells) revealed that they neither exhibit a metaphase delay nor a cytokinesis defect but show errors in chromosome segregation [16]. Thus, it appears that Asp depletion affects mitotic progression in some but not all *Drosophila* cell types.

Defects in mitotic progression have not been detected in most Aspm- and ASPM-depleted cells. In mouse *Aspm* mutants lacking either exons 8–28 or 26–28 of the gene (wild-type *Aspm* contains 28 exons), the NE cells of the embryonic dorsal telencephalon showed relative frequencies of prometaphases/metaphases and anaphases/telophases comparable to those of wild-type cells, indicating that loss of the Aspm function does not result in a detectable defect in mitotic progression [22]. However, Aspm-depleted granule cells of the postnatal mice cerebellum showed an increase in the MI and in the proportion of prometaphase cells compared to the control, suggesting that Aspm is required for mitotic progression [49]. There are no reports suggesting a role of ASPM in the mitotic progression of human cells. RNAi-mediated depletion of ASPM in U2OS cells resulted in a reduction of the MI with no evidence of prometaphase/metaphase delay but caused cytokinesis failures [35]. Furthermore, live cell microscopy of a human HCT116 cell line in which the entire open reading frame of *ASPM* was removed using the CRISPR/Cas9 technology did not reveal any abnormality in mitotic progression [50]. 

### 4.2. The Role of Asp/ASPM Proteins in Chromosome Segregation

The role of *asp* in the control of chromosome segregation was first detected by genetic analysis of sex chromosome segregation in *asp* mutant males [1]. These males produced frequent nullo (0) and diplo gametes (XY and XX), indicating aberrant segregation during both meiotic divisions. Cytological observations further showed that *asp* mutations do not disrupt chromosome pairing at meiosis I but alter chromosome migration towards the poles [1]. Consistent with these results, observations of spermatid nuclei of *asp* mutant males revealed that they have different sizes, indicative of different chromosomal contents [1,24,28,42].

Several results indicate that *asp* is also required for mitotic chromosome segregation. The hyperploid cells observed in larval brains of *asp* mutants [1] suggest that when these cells divide, perhaps in young larvae when the maternal Asp pool is not exhausted, they have defects in chromosome segregation. Consistent with this hypothesis, flies homozygous for *asp* mutations generated by heterozygous mothers showed a substantial increase in cuticular clones derived by the loss of a marked Y chromosome compared to their heterozygous sibs [51]. Abnormal mitotic chromosome segregation was also observed in embryos from *asp* mutant mothers (which lay eggs that undergo some syncytial divisions but do not reach the larval stage) [52]. Moreover, time-lapse recording of live cell division in the neuroepithelium of *asp* mutant larvae showed that 23% of the anaphases exhibit errors in segregation [16].

To the best of our knowledge, no clear defects in chromosome segregation have been described in ASPM- or Aspm-deficient cells. 

### 4.3. The Roles of Asp/ASPM Orthologs at the Spindle Poles and Their Interaction with the Centrosome

Consistent with their localization during cell division, all Asp orthologues control the organization of the spindle poles. In *Drosophila* embryos from *asp* mutant mothers, the centrosomes are frequently dissociated from the spindle poles [52]. Similarly, in central brain neuroblasts of *asp* mutants, spindle poles are often unfocused, and centrosomes are frequently detached from the poles [24]. Moreover, mutant metaphases have fewer and generally shorter astral MTs than their wild-type counterparts. These defects are independent of abnormalities in the centrosomes, as the centrosomes of *asp* mutants recruit normal amounts of γ-tubulin and Cnn, and *asp asterless* (*asl*) double mutants that lack functional centrosomes assemble bipolar spindles with unfocused poles [24]. An analysis of live neuroblasts of *asp* mutants further showed that the centrosomes detached from the poles can move randomly within the dividing cell so that, in some cases, one of the daughter cells inherits both centrosomes [18]. This observation is consistent with an early study reporting that *asp* mutant brains exhibit hemi-spindle-like structures associated with a single centrosomal signal [52]. Furthermore, unfocused spindle poles and centrosome detachment from the poles have been observed in Asp-depleted S2 cells [21,26,48,53]. Poorly focused spindle poles were also observed in live NE cells of *Drosophila asp* mutants [16], as well as in epithelial cells of the fly pupal notum, which also showed centrosome detachment from the poles [25]. Finally, unfocused spindle poles and detached centrosomes have been described in male meiotic cells of *Drosophila asp* mutants, which also showed asters smaller than in the wild type [24]. 

Although Aspm accumulates at the spindle poles in different types of mouse cells, no defects have been reported in spindle pole focusing or aster formation upon Aspm loss [22,32,49]. In addition, no clear defects in spindle poles and asters were described in ASPM-depleted U2OS human cells [35]. In contrast, ASPM-deficient HeLa cells displayed a strong reduction in astral MTs [19,54]. No spindle abnormalities were observed in a human HCT116 cell line missing the entire coding region of the *ASPM* gene [50]. However, these mutant cells responded differently from cells with an intact *ASPM* gene to RNAi against *CDK5RAP2/CEP215*, a *cnn* ortholog responsible for human microcephaly [55]. More than 80% of cells depleted of both ASPM and CDK5RAP2 displayed unfocused spindle poles, which were virtually absent in cells depleted of either ASPM alone or CDK5RAP2 alone [50]. Interestingly, the doubly depleted cells also showed a four-fold increase in prometaphase/metaphase duration compared to both singly depleted cells and normal cells. Thus, ASPM is required for spindle pole focusing and mitotic progression also in HCT116 cells, but this function becomes evident only in the absence of CDK5RAP2. 

An interpretation consistent with the aster and spindle pole phenotypes elicited by the loss of the Asp/ASPM orthologs is that these proteins mediate interactions between the MTs minus ends and the centrosome. The Asp/ASPM orthologous proteins specifically target the MT minus ends but weakly bind also the MT lattice [19]. We suggest that Asp/ASPM proteins accumulated at the spindle poles mediate crosslinking between the MT minus ends and between the minus ends and the sides of other MTs, including those still attached to the centrosomes, ensuring both spindle pole focusing and astral MT stability (Figure 2). In *asp/ASPM* deficient cells, the MTs would be normally nucleated by the centrosome but would not be held in its vicinity after their release from this organelle, leading to reduced aster size and centrosome detachment from the spindle poles (Figure 2). We further suggest that the spindle pole/aster phenotype is not observed in all cell types because, in certain cells, the loss of the Asp/ASPM function is compensated by the activity of another protein, as shown by Tungadi et al. [50]. We note that this model explains why partially purified centrosomes from *Drosophila* embryos require Asp to form aster-like structures in vitro in the presence of rhodamine-labeled tubulin [23]. It is indeed likely that Asp links to the centrosome the MTs that have detached from this organelle, allowing the formation of MT asters (Figure 2).

### 4.4. The Roles of Asp/ASPM Orthologues in the Orientation of Mitotic Divisions

In both *Drosophila* and mammals, spindle orientation plays a critical role in neurogenesis. During early neurogenesis, neural progenitor cells (NPCs) undergo a series of symmetric divisions (with the spindle axis parallel to the neuroepithelium plane), thereby expanding the NPC pool. After increasing their number, NPCs divide asymmetrically (with the spindle axis oblique/orthogonal to the neuroepithelium plane), generating another NPC and a cell that will differentiate into a neuron. Defects in spindle orientation and cell cycle/mitotic delays can result in premature differentiation leading to a reduction in the final neuron number and in brain size [56]. Many studies have shown that astral MTs play a fundamental role in spindle orientation [57,58,59]. Consistent with these studies, the Asp orthologous proteins have been implicated in spindle orientation. A first analysis of the role of Asp/ASPM proteins in spindle orientation has been performed on mouse NE cells. Upon RNAi-mediated *Aspm* knockdown, the spindle axes of NE cells were parallel to the surface of the neuroepithelium less frequently than in the wild type, indicating a precocious switch from proliferative to neurogenic divisions, accompanied by a reduction of the expansion of the NE progenitor pool [32]. This suggested that a deviation of the spindle axis in NE cells could be one of the factors leading to *ASPM*-dependent human microcephaly [32]. However, in a subsequent study from the same group, two mutations leading to truncated Aspm proteins (one missing exons 1–7 and the other only the C terminal exon) failed to cause spindle misorientation in NE cells, although they resulted in mild microcephaly [22]. To explain these conflicting results, it was suggested that the truncated Aspm proteins might partially fulfill the function of the full-length Aspm [22]. Consistent with this interpretation, misoriented spindles were observed in mice bearing a different Aspm truncation mutation, leading to a protein lacking the C terminal 1717 amino acids [60]. Aspm deficiency also produced significant changes in the orientation of divisions within the external granule progenitor layer of mouse postnatal cerebellum, but it is unclear whether these changes contributed to a reduction of neuronal cells [49].

The roles of Asp/ASPM proteins in the control of spindle orientation have also been evaluated in cultured human cells and in the *Drosophila* model. Human tissue culture cells normally divide with the spindle axis parallel to the substratum (e.g., the cell culture dish). However, in U2OS osteosarcoma cells subjected to *ASPM* RNAi, there was a significant increase in spindles with axes perpendicular to the substrate compared to controls [35]. Similarly, ASPM-depleted HeLa cells showed a substantial increase in misoriented spindles compared to non-mutant controls [19,54]. *Drosophila* adults and pharate adults (adults still encased in the puparium) homozygous for *asp* mutations showed a striking reduction of the head size, while the abdomen and thorax were of normal size [16]. To address the mechanism underlying this phenotype, spindle orientation was analyzed by time-lapse recording in the optic lobe neuroepithelium. In early third instar wild-type larvae, most spindles were parallel to the epithelium plane and divided symmetrically. In contrast, in *asp* mutant larvae, there was a premature switch of spindle positioning towards an oblique/perpendicular position, with 50% of the cells undergoing an asymmetrical division [16]. Noteworthy, the small head phenotype was attributed not only to spindle mispositioning and defective chromosome segregation in the neuroepithelium (see Section 4.2 above) but also to an effect of *asp* mutations on myosin-II distribution resulting in defective tissue architecture [16].

### 4.5. The Roles of Asp/ASPM Orthologues in Cytokinesis

The fact that in *Drosophila asp* mutant larvae, the cells of the central brain arrest in metaphase due to the SAC activity prevented the study of the role of Asp in telophase and cytokinesis. However, in male meiotic cells, the SAC is weak and causes only a slight delay of anaphase onset [61]. Consistent with the finding that Asp binds the MT minus ends of the central spindle, in meiotic cells of *asp* mutant males, the central spindle is often disorganized, leading to mislocalization of midbody proteins, defective formation of the actin-based contractile ring and failures in cytokinesis [24,28]. 

Asp-dependent cytokinesis failures were not reported in several studies on S2 tissue culture cells [18,21,26] and in a study on the epithelial cells of *Drosophila* pupal notum [25]. In addition, it was explicitly excluded that *asp* mutations affect the cytokinesis of NE cells of the *Drosophila* optic lobe [16]. Cytokinesis failures were described neither in embryonic NE cells [22,32] nor in postnatal cerebellar neuron progenitors [49] of *Aspm* mutant mice. No cytokinesis defects were described in a study on ASPM-depleted U2OS human cells [33], and the presence of cytokinesis failures was explicitly excluded after the analysis of the mitotic behavior of *ASPM* knockout human HCT116 cells [50]. However, another study reported that the majority of ASPM-depleted U2OS cells fail to complete cytokinesis and that failures in cytokinesis were more common in cells that divide asymmetrically with the spindle axis perpendicular to the substrate [35]. Thus, the requirements of the Asp proteins for cytokinesis appear to be cell type-specific in both *Drosophila* and mammalian systems. The mechanisms underlying Asp-dependent cytokinesis failure have been investigated only in *Drosophila* meiotic cells. In *asp* mutant males, a substantial fraction of telophases displayed central spindles with irregularly arranged MTs and abnormally shaped contractile rings [24,28]. These defects suggested that Asp is required to cross-link the minus ends of the central spindle MTs preventing them from sliding and splaying apart, thus allowing proper overlap of the MT plus ends at the midzone, a precondition for contractile ring formation [24,62] (Figure 2). 

### 4.6. The Role of Asp in Kinetochore-Driven MT Growth

Recent work in S2 cells has shown that Asp plays a role in kinetochore-driven MT regrowth after colcemid-induced MT depolymerization [27]. Asp-GFP showed a peculiar and unexpected behavior during the spindle regrowth process. After colcemid treatment, the spindles of more than 95% of prometaphases and metaphases appeared to be completely depolymerized and did not exhibit detectable remnants of the spindle MTs. Surprisingly, however, about 50% of these cells showed Asp-GFP accumulations near the kinetochores. With the progression of MT regrowth, Asp-GFP is associated with the minus ends of the regrowing MT bundles and eventually accumulated at the poles of reassembled spindles [27]. Notably, Asp negatively regulated the process of spindle reformation, as in Asp-depleted cells, MT regrowth from kinetochores was faster than in controls [27]. A hypothesis to explain these findings is that Asp interacts with either some kinetochore-associated spindle remnants or with one or more kinetochore proteins. Regardless of their mechanistic interpretation, these findings suggest some intriguing speculations. For example, Asp might bind the minus ends of the short MTs that form near the kinetochores and help to establish a correct kinetochore-MT attachment, avoiding a SAC-dependent metaphase delay (see [27,63] for insight into this possibility). It is currently unknown whether the mouse and human Asp orthologs have roles in kinetochore-driven MT growth.

## 5. Asp/ASPM Interacting Proteins

Mass spectrometry and Western blotting experiments showed that Asp binds the essential non-muscle Myosin II light chain, an interaction that is probably mediated by the Asp IQ motifs (Figure 1; [64] and references therein). However, this interaction does not appear to be relevant for mitosis but rather for the maintenance of proper tissue architecture during the morphogenesis of the *Drosophila* neuroepithelium [16]. Another Asp binding partner is calmodulin (CaM), and also, in this case, the interaction is likely to involve the Asp IQ motifs [18]. Interestingly, in both live S2 cells and neuroblasts, Asp and CaM colocalize in discrete puncta along the spindle MTs and move together toward the spindle poles. Moreover, disruption of the Asp-CaM interaction resulted in unfocused spindle poles and centrosome detachment [18,65]. These data suggested that CaM regulates the Asp activity, promoting its MT crosslinking ability at the spindle poles [18]. The Asp-CaM interaction and their cooperation in the control of spindle structure are conserved, as they have been found in *C. elegans*, mouse, and human tissue culture cells [17,19,20].

Co-immunoprecipitation experiments carried out on *Drosophila* embryo extracts showed that Asp copurifies with Polo kinase. Consistent with this finding, Polo kinase efficiently phosphorylates the N terminal segment of the Asp protein. Asp phosphorylation by Polo kinase is not required for Asp accumulation at the spindle poles. However, in vitro experiments with partially purified centrosomes showed that phosphorylation stimulates Asp activity to organize MTs into asters [23].

Studies in *C. elegans* showed that the Asp/ASPM ortholog ASPM-1 biochemically interacts with LIN-5, which shares homology with the human NuMA protein [17]. NuMA has functional features similar to those of Asp, as it binds the MT minus ends, streams poleward along the MTs, and accumulates at the spindle poles where it drives pole focusing, centrosome-spindle coupling, and spindle orientation (reviewed by [38,66]). Mud, the *Drosophila* homolog of NuMA, genetically interacts with Asp, although there is no evidence that the two proteins form a complex [25]. In epithelial cells of the pupal notum, Asp-GFP particles streamed poleward, like in S2 cells and larval neuroblasts, and eventually accumulated at the spindle poles [25]. Cherry-tagged Mud (ChFP-Mud) streamed towards the poles concomitant with Asp-GFP, and *asp* mutations prevented both Mud poleward flow and its accumulation at the spindle poles but did not affect its localization at the cell cortex. Conversely, neither Asp-GFP poleward flow nor its accumulation at the spindle poles was affected by the loss of Mud [25]. In epithelial cells of the notum, Mud deficiency resulted in defective spindle pole focusing and centrosome detachment from metaphase spindles as well as abnormal spindle orientation [25]. However, in Asp-depleted cells, the spindle orientation was normal. This finding suggested that both Asp and Mud contribute to spindle focusing and centrosome-spindle coupling, while spindle positioning is under the control of cortical Mud, which is still present in the absence of Asp [25].

The requirement of Asp for Mud localization is not conserved, as ASPM is not needed for NuMA localization in human cells [19]. In addition, the role of Mud at the spindle poles in epithelial cells seems rather specific, as defects in the spindle poles were not described in larval neuroblasts of *mud* mutants [67]. Moreover, RNAi-mediated Mud depletion did not induce spindle pole disorganization in S2 cells [65,68,69]. These data have led to the suggestion that Asp, and not Mud, is the “functional ortholog” of NuMA [18,21,24,26,70]. However, while NuMA binds to dynein-dynactin and is transported to the spindle poles by this complex [71], Asp does not interact with dynein-dynactin, and its localization to the poles of S2 cells is not dependent on this minus end-directed motor [21]. Asp accumulation to the poles of S2 cells is also independent of the minus end-directed Ncd (HSET in mammals) kinesin-like protein [21,53]. It has been proposed that Asp is transported to the poles by the MT flux [21], a process generated by the MT plus end growth at the kinetochores and minus end depolymerization at the spindle poles [72]. This suggestion is supported by two findings. First, the velocity by which the Asp-GFP particles move poleward is similar to the velocity of tubulin subunits during their flux toward the spindle poles. Second, the depletion of factors that reduce the flux rate also reduces the Asp-GFP motility [21].

ASPM physically associates with the MCPH protein WDR62 (MCPH2) [36]. In HeLa cells, WDR62 accumulates around the centrosomes and at the spindle poles just like ASPM and is required for pole-centrosome coupling, mitotic progression, and spindle orientation [73]. Both ASPM and WDR62 were found to interact with the centrosomal proteins CEP63, responsible for a form of Seckel Syndrome, and with CENPJ/CPAP/Sas-4, which is another MCPH protein (MCPH6) [36]. In mice, Aspm and Wdr62 genetically interact and are thought to be required for proper apical complex structure and localization but not for spindle rotation. It has been suggested that the abnormality of this complex leads to precocious differentiation of neural progenitors leading to a reduction of the neurogenic cell population [36]. 

ASPM physically interacts with Citron kinase (CITK) in both coimmunoprecipitation experiments and in situ proximity ligation assay (PLA) [34,54]. The two proteins largely colocalize at the spindle poles, and CITK recruitment at the poles is significantly reduced in ASPM-depleted cells, while CITK-deficiency does not affect ASPM accumulation at the poles [54]. It has also been observed that CITK and ASPM cooperate in aster formation and stability and that CITK regulates spindle orientation in both the embryonic mouse cerebral cortex and HeLa cells [54]. Interestingly, the role of CITK in spindle orientation is evolutionarily conserved, as the *Drosophila* ortholog of CITK is also required for proper spindle orientation in dividing larval neuroblasts [54]. 

An additional ASPM interacting partner is katanin, an MT-severing complex consisting of the p60 ATPase and the p80 regulatory subunit [19]. Mutations in the human p80-coding gene (*KATNB1*) cause defects in spindle architecture and result in microcephaly [74]; similarly, loss of *Drosophila* Kat80 affects neuroblast division in the optic lobe, resulting in reduced brain size [74]. ASPM and katanin colocalize at the MT minus ends and cooperate to block minus end growth in vitro [19]. In dividing human cells, the ASPM-katanin complex accumulates at the spindle poles and weakly associates with the MT lattice [19]. ASPM and katanin are mutually dependent for their localization, and the knockout of each protein/complex reduced by ~50% the localization of the other. Interestingly, the expression of a point mutant of katanin that cannot bind to ASPM causes defects in spindle orientation, highlighting the importance of the ASPM-katanin interaction [19]. These and other elegant experiments suggested that ASPM and katanin bind and regulate the MT minus ends at the spindle poles, mediating correct spindle focusing, aster formation, and spindle orientation [19].

## 6. The Functional Meaning of Asp/ASPM Localization along the Spindle MTs

As mentioned above, *Drosophila* Asp, in addition to its prominent accumulation at the spindle poles, also localizes along the spindle MTs. A relatively weak Asp signal in the nonpolar region of prometaphase and metaphase spindles has been observed in different *Drosophila* cell types, including larval neuroblasts, epithelial cells, and S2 cells [24,25,26,28]. Moreover, it has been reported that in both S2 cells and epithelial cells, the Asp-GFP particles associated with the spindle MTs stream toward the spindle poles with a velocity comparable to that of the MT flux [18,21,25]. It was further noticed that depletion of the Dgt6 augmin subunit reduces the number of Asp-GFP particles in the body of the metaphase spindles [21]. Because augmin mediates MT nucleation from the side of preexisting MTs [75], this finding suggested that Asp-GFP binds the minus ends of the augmin-dependent MTs and is transported to the pole via the flux [21]. The biological meaning of the poleward Asp movement is unclear. A possibility is that Asp contributes to the stability and proper organization of the non-polar regions of the spindle, although no defects in these regions have been described in Asp-deficient cells. The other possibility is that Asp simply works as a poleward carrier for other spindle proteins. This second alternative is supported by the findings that Asp binds CaM and the NuMA homologue Mud and mediates their transport to the spindle poles [18,25]. ASPM-GFP is weakly enriched along the spindle body of different types of human cells (U2OS, HCT116 and HeLa), but it is currently unknown whether it has the ability to stream toward the spindle poles [19,35,50].

## 7. Asp/ASPM Functions in Interphase Nuclei

In most *Drosophila* studies, the localization of Asp in interphase cells was not addressed. The current results suggest that Asp is enriched in interphase nuclei of some cell types, such as S2 cells, epithelial cells, and possibly male meiotic cells. In contrast, Asp does not seem to localize to the embryo and larval neuroblast nuclei [2,24,25,27]. Studies on human and mouse cells seldom addressed the interphase localization of the Asp orthologues. However, three studies clearly showed that ASPM is enriched in the interphase nuclei of both HeLa and U2OS cells [33,35,76]. In addition, although not explicitly stated by the authors, published photographs indicate that Aspm-GFP localizes to the nuclei of the neural progenitors of the postnatal mouse cerebellum [49].

Consistent with their nuclear localization, both human and mouse ASPM/Aspm have been implicated in DNA repair and replication. Early studies reported that RNAi-mediated depletion of ASPM results in a significant reduction of BRCA1, a protein that binds DNA and participates in the repair of double-strand breaks (DSBs) [33,77]. Conversely, in both human and mouse cells, ASPM/Aspm was selectively and significantly downregulated by X-ray irradiation, with an unknown mechanism [78]. It has also been shown that siRNA-mediated ASPM depletion increases the radiosensitivity of human cells in a p53-independent manner. In ASPM-deficient cells, X-ray irradiation significantly increased both the γ-H2AX foci and chromosome aberrations compared to ASPM-proficient cells, suggesting an interaction between ASPM and DNA repair pathways [79]. A more recent study showed that ASPM-GFP is recruited to UV laser-induced DNA damage stripes and that ASPM physically interacts with both BRCA1 and its E3 ubiquitin ligase HERC2, protecting BRCA1 from proteasomal degradation and promoting efficient homologous recombination-mediated repair of DSBs [76]. Consistent with this finding, ASPM depletion also produced chromosome aberrations and sensitized cells to X-ray-induced chromosome damage [76].

ASPM has also been implicated in DNA replication. Affinity purification experiments showed that FLAG-GFP-ASPM interacts with several proteins involved in DNA replication, including the five subunits of the replication factor complex (RFC1-5), the BRCA2 factor involved in both DSB repair and protection of the replication forks, the MCM5 core subunit of the MCM complex that promotes double-strand DNA unwinding at the replication origins, the TopBP1 topoisomerase, and the RAD9 and RAD17 factors that together with TopBP1 associate to the replication forks, facilitating activation of the ATR-CHK1 kinases [80]. Despite its multiple interactions with the DNA replication machinery, ASPM is dispensable for DNA replication under unperturbed conditions [80]. However, after replication stress by either hydroxyurea or aphidicolin, ASPM shields the nascent DNA strand from MR11-mediated degradation at the stalled replication forks, thus preventing genomic instability [80].

Cyclin E, which is required for the G1-S transition, mostly resides in the nucleus and also mediates the nuclear localization of its binding partner Cdk2 [81]. Coimmunoprecipitation experiments with HEK293T human cell extracts showed that ASPM copurifies with the Cyclin E/Cdk2 complex [60]. Consistent with this finding, assays with neurospheres from embryonic mice revealed that ASPM protects Cyclin E from ubiquitination and proteasome-mediated degradation. In *Aspm* mutant mice, the Cyclin E level was reduced compared to controls, causing a cell cycle delay, particularly at the G1-S transition, and an overall reduction of the cerebral cortex. Restoration of the normal Cyclin E level, or treatment with the proteasome inhibitor MG132, rescued the cell cycle delay in mutant *Aspm* mice but did not correct the abnormal orientation of neurogenic mitoses [60]. Thus, Aspm appears to regulate mouse neurogenesis also through the control of Cyclin E level. 

In summary, it is clear that besides its role in spindle assembly, ASPM/Aspm has additional functions in interphase nuclei of both human and mouse cells, where it interacts with a number of proteins required for DNA replication and/or repair and cell cycle progression. ASPM/Aspm also controls genome stability, especially under stress conditions such as exposure to ionizing radiation or disturbances in DNA replication. It is currently unknown whether *Drosophila* Asp functions in interphase nuclei, an issue that deserves future studies, as current results indicate that fly Asp is a good model to investigate the functions of human ASPM.

## 8. Conclusions and Perspectives

The current results indicate that Asp and its mammalian orthologous proteins exhibit virtually identical localization patterns during cell division. They localize to the spindle poles next to or around the centrosomes and to the outer sides of the central spindle, two spindle regions enriched in MT minus ends. In vitro studies have also shown that Asp/ASPM orthologous proteins preferentially bind the MT minus ends. While the results of the Asp/ASPM localization studies are in good agreement, except for a few conflicting findings discussed earlier (Section 3), the results about the phenotypic consequences of Asp/ASPM depletion appear to vary according to the cell type and organism examined. However, as shown in Table 1, these variations might depend on the particular target of the investigation. For example, some studies were focused on one or two specific phenotypes and did not report about the others, either because they were not pertinent or too weak to be easily appreciated. Yet, other studies explicitly excluded some mitotic phenotypes (Table 1). Collectively the extant investigations on different cell types described seven mitotic processes that are affected by Asp/ASPM depletion: aster formation, spindle pole focusing, centrosome-spindle coupling, spindle orientation, metaphase-to-anaphase progression, chromosome segregation, and cytokinesis. We believe that the processes that regard asters, spindle poles, centrosomes, and spindle rotation are strictly related (see Figure 2), and one might assume that when one of them is affected, the others are more or less disturbed. The Asp-dependent effects on mitotic progression, chromosome segregation, and cytokinesis could be cell/organism-specific. However, it is possible that these effects are either too tenuous to be detected or become visible only in the absence of another mitotic function. A good example of this possibility is provided by the studies on human HCT116 cells lacking the *ASPM* gene [50]. These cells have normal mitotic spindles and mitotic progression, but when depleted of CDK5RAP2/CEP215, whose deficiency causes only minor mitotic defects, show strong defects in spindle pole focusing and a prometaphase/metaphase delay. This suggests that some of the *ASPM* functions that are phenotypically hidden in ASPM-deficient HCT116 cells become evident upon the downregulation of *CDK5RAP2*. These findings and the other current data prompt us to propose a working hypothesis on the mitotic roles of *ASPM* and its *Drosophila* and mouse orthologues. We suggest that these genes play the same conserved mitotic functions in different cell types of *Drosophila,* mouse, and humans and that these functions are phenotypically manifest or hidden depending on the cellular context. Mouse and human Aspm/ASPM also have clear roles in DNA replication and repair. Whether *Drosophila* Asp is involved in these processes has never been addressed. However, we speculate that *Drosophila* Asp might be involved in DNA metabolism just as its mouse and human counterparts. 

Our working hypothesis impacts the models for the etiology of ASPM-dependent microcephaly. As mentioned earlier, MCPH is a genetically heterogeneous disorder caused by mutations in at least 30 different genes ([5,7,8,82]; see also the list of microcephaly genes reported in the OMIM database). These genes control different aspects of chromosome biology and cell division, including chromatin remodeling, chromosome condensation and integrity, DNA damage response, kinetochore-MT attachment, SAC signaling, centrosome duplication and function, spindle formation, and cytokinesis. Strikingly, most, if not all, of these processes are also regulated by *ASPM*, suggesting that mutations in this gene have many diverse inhibitory effects on neural progenitor growth and could be, therefore, particularly effective in hampering neurogenesis. This hypothesis also provides a justification for why *ASPM* is the most frequently mutated gene in human microcephaly.

## Figures and Tables

**Figure 1 cells-12-00922-f001:**
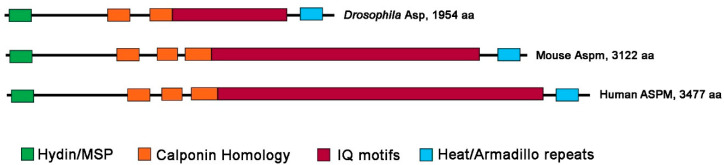
Domain structure of the major isoforms of the *Drosophila* Asp, mouse Aspm, and human ASPM proteins. The N terminal region of the Asp/ASPM proteins contains a Hydin or ASH (ASPM, SPD-2, Hydin) domain, which overlaps with the major sperm protein (MSP) domain. The central part of the proteins contains calponin homology (CH) domains and multiple calmodulin-binding IQ motifs. At the C terminus of the three proteins, there is a conserved Heat/Armadillo repeats domain. (The protein structure analysis is mainly from Rujano et al. [16]).

**Figure 2 cells-12-00922-f002:**
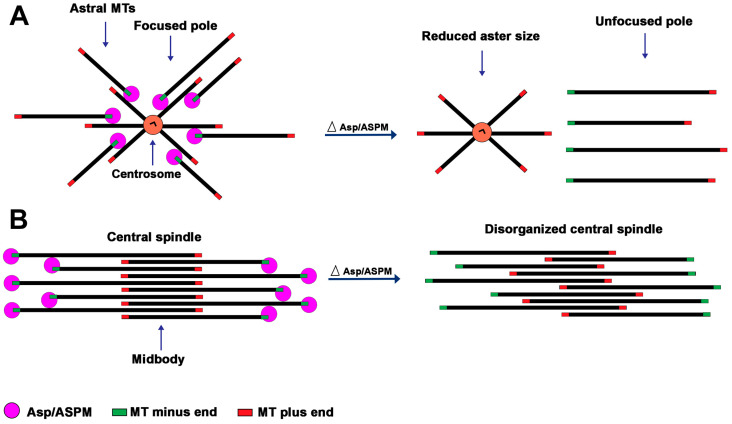
Model for the Asp/ASPM function at the asters/spindle poles and the telophase central spindle. The model has been elaborated based on both the Asp/ASPM localization studies and the phenotypic analyses of the consequences of Asp/ASPM deficiency. (**A**) At the spindle poles, Asp/ASPM binds the MT minus ends, mediating both inter se minus end association (not depicted) and minus end crosslinking to the sides of the MTs attached to the centrosomes, thereby ensuring both spindle pole focusing and astral MT stability. In Asp/ASPM deficient cells, the MTs would be normally nucleated by the centrosome but would not be held in its vicinity after their release from this organelle, leading to reduced aster size, unfocused spindle poles, and centrosome detachment. (**B**) In the central spindle, Asp/ASPM binds the MT minus ends mediating both inter se minus end association (not depicted) and minus end crosslinking to the sides of the other MTs of the bundle, thereby stabilizing the central spindle structure. In the absence of Asp/ASPM, the central spindle MTs would slide along each other, disrupting proper MT plus end overlapping at the midbody, leading to defective contractile ring formation and positioning and cytokinesis failure.

**Table 1 cells-12-00922-t001:** Mitotic processes affected by downregulation of the *asp*/*Aspm*/*ASPM* orthologs.

Organism/Cell Type with Asp/ASPM Depletion	Process Affected						
	Aster Formation	Spindle Poles Focusing	Centrosome Attachment	Spindle Orientation	Metaphase Delay/ Arrest	Chromosome Segregation	Cytokinesis
D.m. central brain neuroblasts (1)	Yes	Yes	Yes	NR	Yes	Yes	N/A
D.m. S2 cells (2)	NR	Yes	Yes	N/A	Yes	NR	NR
D.m. optic lobe neuroepithelium (3)	NR	Yes	NR	Yes	No	Yes	No
D.m. epithelial cells notum (4)	No	Yes	Yes	No	NR	NR	NR
D.m. cuticular cells (5)	N/A	N/A	N/A	N/A	N/A	Yes	N/A
D.m. embryo divisions (6)	NR	NR	Yes	N/A	Yes	Yes	N/A
D.m. male meiotic cells (7)	Yes	Yes	Yes	N/A	N/A	Yes	Yes
Mouse neuroepithelial (NE) cells (8)	NR	NR	Yes	Yes/No	No	NR	NR
Mouse cerebellum cells (9)	NR	NR	NR	Yes	Yes	NR	NR
Human U2OS cells (10)	NR	NR	NR	Yes	No	NR	Yes
Human HeLa cells (11)	Yes	NR	NR	Yes	NR	NR	NR
Human HCT116 cells (12)	NR	No	NR	NR	No	NR	No
Human HCT116 depleted of CDK4RAP2 (12)	NR	Yes	NR	NR	Yes	NR	NR

D.m., *Drosophila melanogaster.* Yes and No, indicate phenotypes explicitly observed or excluded, respectively; Yes/No, indicate conflicting results with the phenotype observed in some studies but excluded in others. NR, phenotype not recorded, namely, the presence/absence of a particular phenotype was either not specifically stated or not considered; N/A, Does not apply. If in multiple studies on the same cell type/organism, a particular phenotype was NR in one/some cases and explicitly observed (Yes) or excluded (No) in other case(s), we reported Yes or No in the table, respectively. Numbers between brackets refer to references: (1) [1,18,24,44,46]. (2) [21,26,53]. (3) [16]. (4) [25]; it has been suggested that the spindle pole phenotype is Mud-dependent (see text for details). (5) [51]. (6) [52]. (7) [1,24,28,42]. (8) Abnormal spindle pole focusing was observed in Fish et al., 2006 [32]. Defects in spindle rotation were observed in Fish et al. [32] and Capecchi and Pozner [60] but not in Pulvers et al. [22] and Jayaraman et al. [36]. (9) [49]. (10) [35]. (11) [19,50,54]. (12) [50].

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
