# Peer review of "The Multiple Mitotic Roles of the ASPM Orthologous Proteins: Insight into the Etiology of ASPM-Dependent Microcephaly"

_cells, 2023, doi:10.3390/cells12060922_

Round 1
Reviewer 1 Report
This manuscript provides an expert review on the molecular cell biology of orthologs of the Abnormal Spindle-like Microcephaly-associated (ASPM) protein family. From the discovery of Asp in Drosophila melanogaster, the ASPM family has come into the focus of biomedical research by its causal association with autosomal recessive primary microcephaly. This review is a well-written, comprehensive review that raises interesting novel concepts in ASPM functions in human disease.
major comments:
(1) The abstract is way too long and thus does not serve its major goal - to attract interested readership to read the full article. My sense is that the abstract could be readily shortened and focussed by eliminating details such as longish listings of individual requirements for ASP/ASPM proteins, and a full spelling out of their working hypothesis.
(2) on page 10 a long paragraph deals with the role of Asp in Kinetochor-driven microtubule growth. While this aspect of Asp functions is still rather unexplored and thus interesting, it is striking that there appears to be only one reference for this. Noteably this reference is co-authored by many of the authors of this review. I would recommend to shorten this part of the manuscript as it is unclear to me why it should be given such a large attention to a single study, when comparing to other much better studied aspects of Asp function.
minor comments:
page 1: typo in the abstract: a t is missing in 'orhologous'
page 2: at the end of the first paragraph there are two sentences in italics. Why is that?
page 15: last paragraph contains again in part italics formatting.
Author Response
We thank the reviewer for the comments. We followed the suggestions and shortened both the abstract (from 295 to 224 words) and the paragraph dealing with the role of Asp in kinetochore-driven microtubule growth (from 339 to 212 words).
Reviewer 2 Report
This review deals with the multiple mitotic role of the the D.m. asp gene and its mammalian orthologs, with a focus on their implication in the etiology of the human microcephaly.
The review is very well written. The literature coverage is exhaustive and in-depth analyzed and discussed. The presence of figures illustrating the models, and tables resuming data, make easy the reading and to follow the Authors’reasoning.
A point the Authors should clarify is related to the role they propose for Asp in the kinetochore-driven MT growth. Asp depletion after colcemid treatment, causes MT regrowth from kinetochores being faster and more abundant than in controls. I suggest that the Authors better explain why must be relevant for a cell to limit the growth of the MT minus ends.
A second point is just a curiosity and not a criticism. The review mainly deals with the mitotic functions of Asp/Aspm/ASPM proteins, but it also encompasses some aspects of their roles in interphase. In this context, I like to ask whether it is known a possible role for these proteins in axon elongation and/or function, which, as it is well known, depend on MT growth.
Some minor editing errors:
Page 5: the title
4. The mitotic functions the asp/ASPM genes should be read: 4. The mitotic functions of the asp/ASPM genes
Page 15: in the sentence “see also the list of microcephaly genes reported in the OMIM database)” the symbol of open parenthesis is clearly missing.
Author Response
We thank the reviewer for the comments. We shortened and tried to simplify the paragraph dealing with the role of Asp in kinetochore-driven microtubule growth. We also added that insight into the possible role of Asp at kinetochores can be found in refs [27] and [63].
As to the reviewer's curiosity, to the best of our knowledge there are no published papers that suggest a direct role of Asp/Aspm/ASPM in axon elongation. There is a study that describes a general nerve fiber alteration in Aspm mutant mice, but it is unclear whether these alterations are a direct effect of lack of Aspm-microtubule interaction (10.1016/j.neuroscience.2017.12.012).